# How Iterative Magnitude Pruning Discovers Local Receptive Fields in Fully Connected Neural Networks

William T. Redman[1,2], Zhangyang Wang[3], Alessandro Ingrosso[4,5], and Sebastian Goldt[6]

[1] UC Santa Barbara, [2] Johns Hopkins Applied Physics Lab, [3] UT Austin,
[4] International Centre for Theoretical Physics, Trieste, Italy, [5] Radboud University,
[6] International School of Advanced Studies, Trieste, Italy.

`will.redman@jhuapl.edu`

Since its use in the Lottery Ticket Hypothesis, iterative magnitude pruning (IMP) has become a popular method for extracting sparse subnetworks that can be trained to high performance. Despite its success, the mechanism that drives the success of IMP remains unclear. One possibility is that IMP is capable of extracting subnetworks with good inductive biases that facilitate performance. Supporting this idea, recent work showed that applying IMP to fully connected neural networks (FCNs) leads to the emergence of local receptive fields (RFs), a feature of mammalian visual cortex and convolutional neural networks that facilitates image processing. However, it remains unclear why IMP would uncover localised features in the first place. Inspired by results showing that training on synthetic images with highly non-Gaussian statistics (e.g., sharp edges) is sufficient to drive the emergence of local RFs in FCNs, we hypothesize that IMP iteratively increases the non-Gaussian statistics of FCN representations, creating a feedback loop that enhances localization. Here, we demonstrate first that non-Gaussian input statistics are indeed necessary for IMP to discover localized RFs. We then develop a new method for measuring the effect of individual weights on the statistics of the FCN representations ("cavity method"), which allows us to show that IMP systematically increases the non-Gaussianity of pre-activations, leading to the formation of localised RFs. Our work, which is the first to study the effect of IMP on the statistics of the representations of neural networks, sheds parsimonious light on one way in which IMP can drive the formation of strong inductive biases.

## 1. Introduction

Iterative magnitude pruning (IMP) [1] has emerged as a powerful tool for identifying sparse subnetworks ("winning tickets") that can be trained to perform as well as the dense neural network model from which they are extracted [2, 3]. That IMP, despite its simplicity, is more robust in discovering such winning tickets than more complex pruning schemes [4] suggests that its iterative coarse-graining [5] is especially capable of extracting and maintaining strong inductive biases. This perspective is strengthened by observations that winning tickets discovered by IMP are transferable across related tasks [6–13] and architectures [14]; that they can outperform dense models on classes with limited data [15]; and that they are less prone to overconfident predictions [16].

The first *direct* evidence for IMP discovering good inductive biases came from Pellegrini and Biroli [17], who found that the sparse subnetworks extracted by IMP from fully-connected neural networks (FCNs) had local receptive fields (RF) (Fig. 1A). Localised RFs are well-suited for image processing, and are also found in the visual cortex [18, 19] and in convolutional neural networks (CNNs) [20, 21]. Comparing the sparse subnetworks found by IMP and oneshot pruning (Fig. 1B), Pellegrini and Biroli [17] argued that the iterative nature of IMP was essential for refining the local RF structure. However, an understanding of how IMP, a pruning method based purely on the magnitude of the network parameters, is able to yield localised receptive fields remains unknown.

Second Conference on Parsimony and Learning (CPAL 2025).

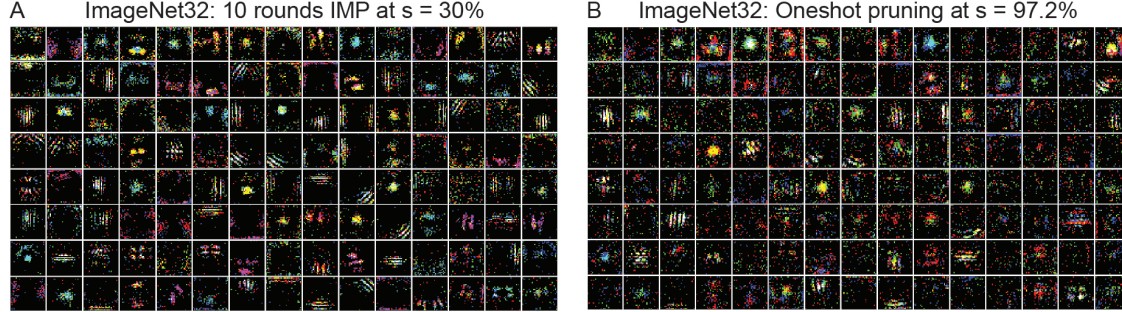

Figure 1: **IMP discovers more localized RFs than oneshot magnitude pruning in FCNs.** (A) Localized RFs are present after applying IMP for 10 rounds of pruning (each round pruning $s = 30\%$ of the remaining weights), leading to a subnetwork with $s = 97.2\%$. (B) Noisier, less localized RFs are present in the masks found after oneshot pruning FCNs trained on ImageNet32 to $s = 97.2\%$ sparsity. Pruned weights are shown in black and remaining weights are colored by which input channel (red, green, blue) they are connected to. The masks shown correspond to the 120 hidden units with the greatest number of weights remaining [17].

Historically, the study of local RFs has focused on specific features of natural images (e.g., sharp edges), and it has been shown that, with regularization, it is possible for local RFs to emerge in FCNs [22–25]. Recent work has built upon this, showing that synthetic images with sufficiently strong non-Gaussian statistics can, without any regularization, drive the formation of local RFs in FCNs [26, 27]. Inspired by this, we hypothesize that IMP (Sec. 2.1) is able to discover local RFs (Sec. 2.2) in FCNs by iteratively increasing the non-Gaussian statistics present in its internal representation. This could create a feedback loop, where the amplification of non-Gaussian statistics leads to greater localization, which leads to further increases in non-Gaussian statistics. Because of IMP's broad success in computer vision, we hypothesize that IMP *maximally* increases the non-Gaussian statistics, by removing exactly the weights that maximize the "non-Gaussanity" of the internal representations. We formalize our hypothesis below (with kurtosis and pre-activations defined precisely in Sec. 3).

**Hypothesis (H∗):** *IMP discovers local RFs in FCNs by maximally increasing the kurtosis of the network's preactivations.*

While this is a challenging hypothesis to prove, we provide the following evidence in support of it:

- Training FCNs on a "Gaussian clone" [28] that matches the original dataset's first two cumulants (mean and covariance), while containing no higher-order cumulants, we find that IMP does not discover localized RFs. This demonstrates that non-Gaussian statistics are *necessary* for IMP to discover local RFs in FCNs (Sec. 3.1).

- Measuring the non-Gaussian statistics present in the preactivations of the FCN, we find that IMP leads to representations with *more strongly* non-Gaussian statistics, as compared to oneshot pruning (Sec. 3.2). This difference increases with each round of pruning, demonstrating the importance of the iterative nature of IMP.

- Developing a method that measures the effect individual weights have on the statistics of the FCN representations, we find that IMP removes weights *precisely* when their pruning would increase the non-Gaussanity of the FCN representations (Sec. 3.3). This suggests that IMP not just increases, but maximizes, the non-Gaussian statistics at each round of pruning.

Collectively, our work provides the first in-depth analysis of how IMP affects the statistics of the internal representations of DNNs, as well as provides insight on how these statistics interact with the iteratively evolving architecture achieved by IMP. Additionally, it highlights the importance of going beyond the assumption of Gaussian features that dominates the current theory of machine learning [29–35]. Our work motivates further study of how IMP learns parsimonious structure in DNNs, and provides tools (Gaussian data clones and the cavity method) to achieve this goal.

## 2. Background

### 2.1. Iterative magnitude pruning

Given a neural network $f(\theta; \mathcal{X})$, where $\theta$ are its $N$ parameters (e.g. its weights, biases) and $\mathcal{X}$ is a set of data samples used for training, IMP [1, 2] performs the following iterative pruning procedure. First, $f(\theta, \mathcal{X})$ is trained for $T$ iterations, resulting in $f[\theta(T); \mathcal{X}]$. Then, a mask $m \in \{0, 1\}^N$ is computed by assigning $m_i = 0$ if the magnitude of the $i$th weight $\theta_i(T)$ is smaller than some fixed threshold $\tau > 0$. For network parameters with magnitude greater than $\tau$, $m_i = 1$. Typically, $\tau$ is set such that $s = 1 - \frac{1}{N} \sum_{i=1}^{N} m_i$, for a desired sparsity $s \in (0, 1]$. Having computed the mask, the network parameter values are "rewound" [36, 37] to a previous value, $\theta(t_{\text{rewind}})$, where $t_{\text{rewind}} \ll T$. The original work leveraging IMP to discover winning tickets demonstrated that winning tickets could be found at initialization (i.e. $t_{\text{rewind}} = 0$). Subsequent work on has found it necessary to set $t_{\text{rewind}} > 0$ [3], particularly when considering architectures larger than LeNet [21] and datasets more complex than MNIST. A common approach is to set $t_{\text{rewind}} \approx 0.01T$. The network $f[\theta(t_{\text{rewind}}) \odot m; \mathcal{X}]$ is then trained for $T - t_{\text{rewind}}$ training iterations, where $\odot$ denotes element-wise multiplication.

This train, prune, and rewind procedure is then repeated, with round $n$ of IMP involving training $f[\theta(t_{\text{rewind}}) \odot m(n-1); \mathcal{X}]$ and computing the mask $m(n) \in \{0, 1\}^N$ from the remaining (non-pruned) parameters $\theta(T) \odot m(n-1)$. The masks $m(n)$ are non-trainable parameters, and $\sum_{i=1}^{N} m_i(n) < \sum_{i=1}^{N} m_i(n-1)$. This is repeated for $N_{\text{IMP}}$ rounds.

IMP has been used to discover sparse subnetworks that can be trained to good performance across a wide range of architectures and tasks [3, 6, 7, 11, 12, 14, 38–42], demonstrating its robustness. Work studying the effectiveness of IMP has focused on the associated loss landscapes[3, 43–45], which has provided evidence supporting the hypothesis that IMP derived subnetworks converge to similar solutions as the full, dense model. Connections between IMP and the renormalization group, a tool used in statistical physics to extract the "relevant" degrees of freedom [46], have been made, allowing for interpretation of "universality" of winning tickets across tasks [5]. The interplay between IMP and the amount of data used to train DNNs has been explored [45, 47, 48], with IMP being more successful when the intrinsic dimensionality of the data is lower.

To the best of our knowledge, our work is the first to analyze the effect of IMP on the statistics of the internal representations in neural networks.

### 2.2. Local receptive fields

**Early work on local receptive fields**  Local RFs were first discovered in mammalian primary visual cortex (V1) [18, 19, 49], where individual V1 neurons were found to respond to specific visual features (e.g. the presence of lines) in a local range of visual space. The computational paradigm of breaking input space into local patches, over which a hierarchical representation could be learned, inspired the development of CNNs [21, 50]. Early work studying how local RFs might emerge in artificial neural networks showed that $L_1$ regularization of hidden unit activations was capable of driving localization in FCNs [22–25]. Thus, while sparsity in activations was known to lead to local RFs, it was not until subsequent effort using a variant of LASSO regression ($\beta$-LASSO) on network parameters that it was appreciated that sparsity in the weights could also lead to localized RFs [51]. Unlike this prior literature, our work focuses on the emergence of local RFs without explicit regularization [17, 26, 27].

**Quantifying the locality of receptive fields**  To quantify the locality of the IMP masks, $m(n)$, we use the following metric. Let $X \in \mathcal{X}$, be a square image with $N_p^2 \in \mathcal{N}$ pixels ($N_p$ in the $x-$dimension and $N_p$ in the $y-$dimension). Let the locations of two pixels of $X$ be denoted as $z = (x, y)$ and $z' = (x', y')$, where $z, z' \in [1, ..., N_p] \times [1, ..., N_p]$. Let the relative position of $z$ and $z'$ be denoted as $d_{zz'} = z - z'$. We define the number of pixels in $X$, with relative position $d$, that are connected to

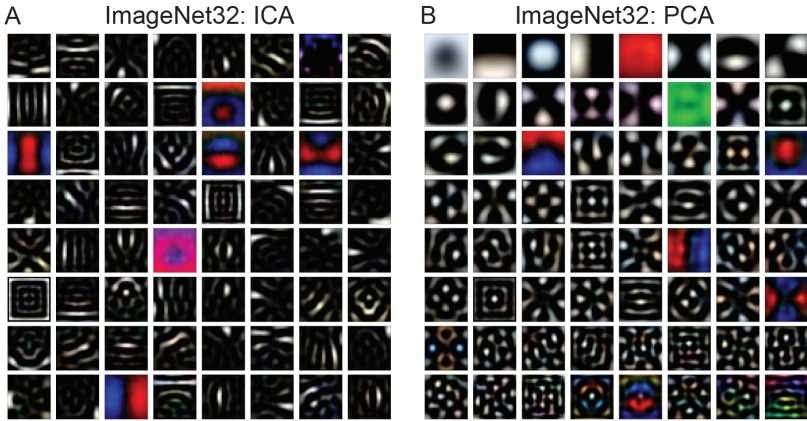

Figure 2: **Non-Gaussian statistics contain local information in ImageNet32.** (A) By maximizing the non-Gaussanity of a lower dimensional representation of the 50,000 validation images from ImageNet32, ICA extracts features, some of which are localized. (B) In contrast, considering only the covariance of the validation images from ImageNet32 leads PCA to extract features that are periodic and thus, non-local.

unit $i$ in the first hidden layer of an FCN, at IMP round $n$, by

$$S_i(d, n) = \sum_z \sum_{z'} \delta(d_{zz'} - d) \cdot m_{iz}(n) \cdot m_{iz'}(n), \tag{1}$$

where $\delta(d_{zz'} - d)$ is a delta function that is equal to 1 when $d_{zz'} = d$, and 0 otherwise. $S_i(d, n)$ acts as a correlation function on the mask associated with unit $i$, $m_i(n)$. Pelligrini and Biroli (2022) [17] previously used $S(d, n)$ to visualize the locality of the IMP masks. We further use it to quantify the localization by computing the standard deviations, $\sigma_x$ and $\sigma_y$, of a two-dimensional Gaussian fit to $S_i(d, n)$ (Fig. S1A). These standard deviations act as a measure of correlation length, with the smaller $\sigma_x$ and $\sigma_y$ are, the more localized the RFs of the IMP masks are. We report $\sigma_x$ as the RF width (similar results are found with $\sigma_y$). Note that, because the Gaussian fits are applied to this correlation function, $S_i(d, n)$, it can be appropriate and capture aspects of local RF width, even in cases where IMP masks do not appear to be Gaussian (Fig. S1). For more details, see Appendix A.

## 3. Results

### 3.1. Non-Gaussian statistics are necessary for IMP to discover local RFs in FCNs

**Key statistical properties of images are encoded in their non-Gaussian statistics.** A key property of natural images is the presence of sharp changes in luminosity, for example at the edges of objects. Edges have long been recognized as a hallmark of natural images [22–25]. Statistically speaking, these edges manifest themselves at the level of higher-order cumulants (HOCs), which capture the correlations between groups of three or more pixels that cannot be decomposed into products of correlations between smaller groups of pixels. For example, the fourth-order cumulant captures four-pixel correlations that cannot be decomposed into products of pair-wise correlations. HOCs are intrinsically non-Gaussian: in a Gaussian distribution, all HOCs are equal to zero. Hence some of the key features of images, such as the presence of sharp edges, cannot be reproduced using a Gaussian model for images here all cumulants after the second are zero. Seminal work on learning local RFs has emphasized the importance of these non-Gaussian statistics for unsupervised learning [22–25], while more recent work demonstrated that synthetic images with strongly non-Gaussian statistics can be sufficient to drive local RF formation in FCNs trained on a supervised classification task [26, 27].

**The most non-Gaussian projections of images are localised** To illustrate why non-Gaussian statistics play a crucial role in the learning of local RFs, we perform independent component analysis

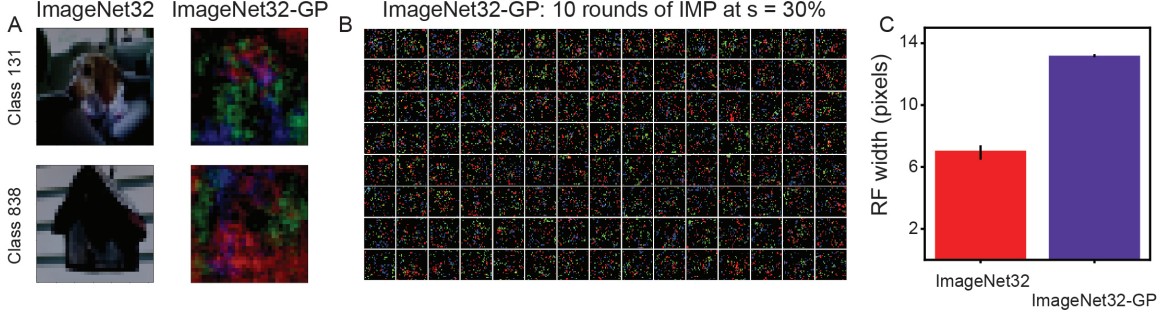

Figure 3: **IMP does not discover local RFs when applied to FCNs trained on a Gaussian clone of ImageNet32.** (A) Example images from ImageNet32 and ImageNet32-GP. Note the lack of sharp edges in the case of ImageNet32-GP. (B) Pruning mask found after 10 rounds of IMP. Compare these diffuse masks with the localized masks found on ImageNet32 (Fig. 1A). (C) Median RF width for masks found by IMP on ImageNet32 and ImageNet32-GP. The smaller the width, the more localized the mask. Error bars are minimum and maximum of three independently trained and pruned FCNs.

(ICA) [52] and principal component analysis (PCA) on images from the ImageNet32 dataset [53], a downsampled version of the original ImageNet dataset (see Appendix B for more details). ICA looks for linear projections, $\lambda = W \cdot X$, which maximize the "non-Gaussianity" of the features $\lambda$. Running this algorithm on ImageNet32 leads to components $W$ that have largely localized features (Fig. 2A). In contrast, running PCA, which only acts on the covariance of the inputs (and thus, only has access to the Gaussian features of the data), yields $W$ with oscillating components (Fig. 2B). Similar results are found when applying ICA and PCA to the full ImageNet dataset. These oscillations provide a spatially distributed and thus, non-local, representation of the images. Therefore, local information is carried primarily by the non-Gaussian statistics of natural images. While both the IMP masks and the ICA components are localized, the exact features they discover differ (Fig. S2). This suggests that IMP is not providing an approximation to ICA. This could be due to the fact that IMP is applied to FCNs trained on an image classification task, as opposed to ICA, which is applied as an image reconstruction task.

**IMP does not discover local RFs when applied to FCNs trained on data with only Gaussian statistics.** Given the results from Fig. 2, we hypothesize that non-Gaussian statistics are *necessary* for IMP to find local RFs in FCNs. To test this, we generate a Gaussian clone of ImageNet32 (ImageNet32 sampled from a Gaussian process – "ImageNet32-GP") [28]. This dataset matches the mean and covariance of ImageNet32, but is explicitly constructed to not contain any higher-order cumulants (Fig. 3A – see Appendix C.1, for details). Unlike when IMP is applied to FCNs trained on ImageNet32, IMP fails to find local RFs in the ImageNet32-GP setting. The masks contain no obvious structure (Fig. 3B) and our metric for localization (Sec. 2.2) shows that the masks found on ImageNet32-GP have a significantly larger correlation length than the masks found on ImageNet32. These results demonstrate that non-Gaussian statistics, in the form of higher-order cumulants, are necessary for local RFs to emerge via IMP.

## 3.2. IMP increases non-Gaussian statistics of FCN representations

Because the non-Gaussian statistics present in natural images contain information that is localized, a neural network that develops local RFs should increasingly represent these non-Gaussian statistics [26]. To quantitatively probe this, we can measure the "non-Gaussanity" of a given neural network's representations. One such way of doing this is by computing the excess kurtosis of the pre-activations, as we describe below.

**The pre-activations of fully-connected neural networks** The activation of units in the first hidden layer of the neural network, $f(\theta; \mathcal{X})$, for an image $X \in \mathcal{X}$, are given by

$$a_i(X) = \sigma \left( \sum_{j=1}^{N_p^2} W_{ij} X_j + b_i \right), \tag{2}$$

where $\sigma(\cdot)$ is a nonlinear function (e.g., ReLU), $W_{ij}$ is the weight of pixel $j$ to hidden unit $i$ (in the first layer), and $b_i$ is the bias of unit $i$. The "preactivation" of the $i^{\text{th}}$ unit (i.e., the value of the hidden unit before the application of the activation function), for a given input $X$, is thus defined as

$$\lambda_i(X) = \sum_{j=1}^{N_p^2} W_{ij} X_j + b_i. \tag{3}$$

In cases where batch normalization [54] is used, the input is transformed, becoming $\tilde{X}_j = \gamma_j X_j + \beta_j$, where $\gamma_j$ and $\beta_j$ are parameters that are learned during the course of training. In this case, Eq. 3 is modified such that $\lambda_i(\tilde{X}) = \sum_{j=1}^{N_p^2} W_{ij} \tilde{X}_j + b_i$.

**Measuring non-Gaussian statistics in neural networks representations.** At initialization, the weights of the network are drawn i.i.d. from a uniform distribution. The preactivations, $\lambda$, therefore follow a normal distribution, by the Central Limit Theorem. To quantify how the statistics of $\lambda$ evolve during training, the kurtosis can be measured, which is defined as

$$\text{kurt}(\lambda_i) = \frac{\mathbb{E}_{\mathcal{X}} \left[ \lambda_i(X) - \mathbb{E}_{\mathcal{X}} \lambda_i(X) \right]^4}{\left( \left[ \lambda_i(X) - \mathbb{E}_{\mathcal{X}} \lambda_i(X) \right]^2 \right)^2}, \tag{4}$$

where $\mathbb{E}_X$ is the expectation computed over all the inputs $X \in \mathcal{X}$. The kurtosis allows us to quantify how non-Gaussian the distribution of preactivations is. If $\lambda_i$ is distributed according to a Gaussian, then $\text{kurt}(\lambda_i) = 3$. If $\text{kurt}(\lambda_i) > 3$, then the distribution is more peaked than a Gaussian. If $\text{kurt}[\lambda_i] < 3$, the distribution is more broad than a Gaussian. The extent to which a distribution of preactivations is non-Gaussian can measured by its excess kurtosis, defined as $|3 - \text{kurt}(\lambda_i)|$. We chose to use the kurtosis (as opposed to another metric), as it was previously shown by Ingrosso and Goldt (2022) [26] that the kurtosis significantly increased with training as FCNs developed localized RFs.

**IMP increases FCN preactivation kurtosis, relative to oneshot pruning.** Given that IMP leads to more localized RFs than oneshot pruning (Fig. 1) [17], we hypothesize that IMP increases the non-Gaussian statistics being represented by the FCN more strongly than oneshot pruning. In particular, because IMP is applied iteratively, we hypothesize that each round of pruning leads its starting point, $f[\theta(t_{\text{rewind}}) \odot m(n-1); \mathcal{X}]$, to have a greater representation of non-Gaussian statistics, providing a kind of feedback loop that drives the emergence of more localized RFs.

To test this, we compute the kurtosis of the preactivations of $f[\theta(t_{\text{rewind}}) \odot m(n-1); \mathcal{X}]$, for the masks found by both IMP and oneshot pruning. We also compare with a random pruning baseline, to identify how much the observed change in kurtosis is due sparsity alone. As expected, we find that both IMP and oneshot pruning discover masks with significantly more localized structure than random pruning (Fig. 4A). In-line with non-Gaussian statistics driving localization, we find significant increases in the kurtosis of the preactivations only for IMP and oneshot pruning (Fig. 4B). This demonstrates that the increase in local non-Gaussian statistics is not merely a product of sparsification, as random pruning does not see this effect.

When comparing IMP and oneshot pruning, we find that IMP leads to greater preactivation kurtosis (Fig. 4B), at a sparsity that precedes IMP's greater localization (Fig. 4A). In addition, we find that the preactivation kurtosis increases almost monotonically for IMP. This supports our hypothesis that IMP increases the non-Gaussanity of its representations with each round of pruning, to a greater extent than would be achieved by oneshot pruning. Thus, the iterative nature of IMP is playing an important role in amplifying the non-Gaussian statistics.

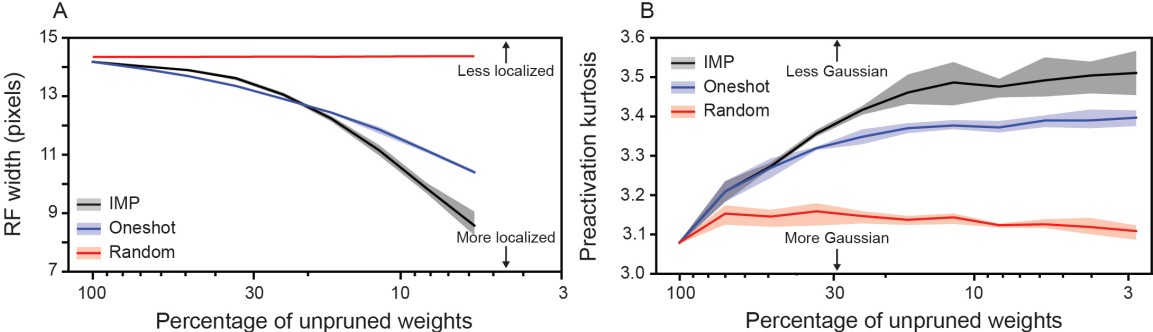

Figure 4: **IMP increases localization of RFs and preactivation kurtosis in FCNs trained on ImageNet32, to a greater extent than oneshot pruning.** (A) Mean RF width, as a function of sparsity induced by IMP (black line), oneshot pruning (blue line), or random pruning (red line). (B) Mean kurtosis of preactivations, per class, as a function of sparsity induced by IMP, oneshot pruning, and random pruning. Note that a kurtosis > 3 implies more non-Gaussian statistics. In (A)-(B), solid line is mean and shaded area is minimum and maximum of three independently trained and pruned FCNs.

Pellegrini and Biroli [17] also found localization of the masks in the later layers of their FCN. The interpretation of this localization is more nuanced, as the hidden units in the later layers are not directly connected to the inputs. However, if non-Gaussianity is a key to localization, then we expect the preactivations in all layers that exhibit localization to become more non-Gaussian across rounds of IMP. Indeed, we find that the preactivation kurtosis in the second layer of our FCN also increases with IMP round (Fig. S3), demonstrating that IMP is also able to increase the non-Gaussianity in later FCN layers.

### 3.3. IMP maximally increases non-Gaussian statistics of FCN representations

**Probing the role of individual weights on the statistics of FCN representations.** While the results presented in Fig. 4 demonstrate that the sparse subnetworks IMP finds after each round of pruning, $f[\theta(t_{\text{rewind}}) \odot m(n-1); \mathcal{X}]$, start with increasingly non-Gaussian representations, as compared to oneshot pruning, they do not provide insight on whether the parameters IMP removes are "optimal" in driving the largest increase in non-Gaussanity. Evidence for this would directly support our central hypothesis (**H**$^*$).

Because IMP removes multiple parameters at once, identifying the choice that maximizes the non-Gaussian representation is intractable due to its combinatorial complexity. We can nonetheless approach testing this hypothesis by considering a simplified setting. In particular, we take inspiration from the "cavity method" of statistical physics [55], and evaluate the impact a given weight in the first hidden layer, $W_{ij}$, has on the statistics of the preactivations $\lambda_i$ by computing the kurtosis of the preactivations, with and without the weight[1]. To quantify weight $W_{ij}$'s impact on the statistics, we develop a metric we call the "cavity score", computed as

$$\text{cavity}(W_{ij}) = \begin{cases} \left(\text{kurt}[\lambda_i^{(-j)}] - \text{kurt}[\lambda_i]\right) / \text{kurt}(\lambda_i) & \text{if } \text{kurt}(\lambda_i) > 3 \\ \\ \left(\text{kurt}[\lambda_i] - \text{kurt}[\lambda_i^{(-j)}]\right) / \text{kurt}(\lambda_i) & \text{if } \text{kurt}(\lambda_i) < 3, \end{cases} \tag{5}$$

where the preactivation with $W_{ij}$ removed, $\lambda_i^{(-j)}(x)$, is defined as,

$$\lambda_i^{(-j)}(X) = \sum_{k=1}^{N_p^2} W_{ik} X_k - W_{ij} X_j. \tag{6}$$

---

[1]We consider only weights here, as removing the biases does not change the kurtosis of the preactivations, as it is just corresponds to a translation of the distribution.

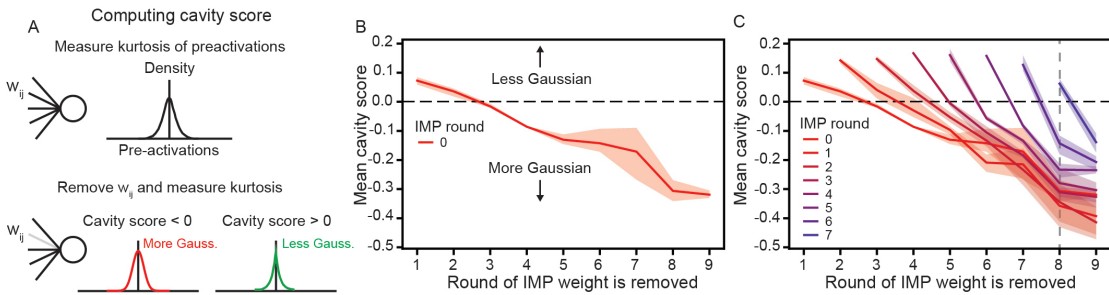

Figure 5: **IMP selectively prunes weights when their removal would most increase the non-Gaussianity of the preactivations.** (A) A schematic overview of how the cavity score (Eq. 5) is computed. For a given unit in the first hidden layer, the kurtosis of its preactivation is computed (top). Then, the preactivation kurtosis is recomputed, with a given weight $W_{ij}$ removed (bottom). If the distribution of preactivations becomes more Gaussian once $W_{ij}$ is removed, cavity($W_{ij}$) $< 0$. If the distribution of preactivations instead becomes less Gaussian, cavity($W_{ij}$) $> 0$. (B) Mean cavity score, computed at IMP round 0, for weights grouped according to the round of IMP they ultimately get removed during. Note that the weights that get removed later during IMP have negative cavity score, while weights that get removed early during IMP have positive cavity score. (C) Same as (B), but when computing the cavity score of the remaining weights, $\theta(t_{\text{rewind}}) \odot m(n-1)$, after each round of IMP. Gray dashed line is used to highlight the fact that the mean cavity score of the weights that get removed at IMP round 8 is negative for all rounds of IMP, until after the 7th round of pruning. In (B)-(C), solid line denote mean, and shaded area is minimum and maximum of three independently trained and pruned FCNs.

A negative cavity score signifies that removing $W_{ij}$ decreases the excess kurtosis of the preactivations (making them more Gaussian), and a positive cavity score signifies that removing $W_{ij}$ increases the excess kurtosis of the preactivations (making them less Gaussian). A schematic of the approach is shown in Fig. 5A.

To get the most accurate estimate of the cavity score, we compute Eq. 5 using the entire ImageNet32 test set. However, this is computationally expensive. To reduce this cost, we could compute the cavity score over a subset of the test set. In particular, we could reduce the number of classes used to compute the cavity score (i.e. consider the average kurtosis across 10 classes of images, instead of 1000). This would accelerate the computation, as currently our method has to recompute the cavity score for each class separately. Future work could explore the trade-off between accuracy of computed cavity score and the number of test data points or classes used. We imagine that a large reduction can be achieved, while still maintaining accurate cavity score computations.

**The order in which IMP prunes maximizes the representation of non-Gaussian statistics in FCNs.** We use the cavity score to probe whether the ordering with which IMP removes weights selectively shapes the preactivation excess kurtosis to be less Gaussian. Re-analyzing the IMP experiments used in Fig. 4, we compute the cavity score for all weights, based on their value at the rewind point, $\theta(t_{\text{rewind}})$ (before any pruning has been performed, i.e. IMP round 0). We then group the weights by which round of IMP they ultimately get pruned during[2]. Plotting the average cavity score, for each grouping, we see that the weights that get removed in the first and second round of IMP have positive cavity score, while the weights that get removed during IMP round 4 or later have negative cavity score (with the weights that get removed at IMP round 3 having cavity score approximately equal to zero) (Fig. 5B). Thus, for a weight that gets removed during the later rounds of IMP ($n \geq 4$), its removal at this initial stage would decrease the non-Gaussianity of the preactivations. In contrast, for a weight that is removed during the earliest rounds of IMP, its removal at this initial stage would increase the non-Gaussanity of the preactivations. We do not find that parameter sign or magnitude is strongly connected to cavity score (Fig. S4).

---

[2]The re-analysis allows us to do this, as we have already run the experiment, and therefore we know the "fate" of each weight.

We then consider whether this same trend holds for the subsequent rounds of IMP. Given that the preactviation distribution for each hidden unit changes after pruning (as some weights get removed), we recompute the cavity score for all the remaining weights after the first round of IMP (at the start of IMP round $n = 2$). As before, the preactivations are measured from the rewind point (now determined by $f[\theta(t_{\text{rewind}}) \odot m(1); \mathcal{X}]$). We again group the remaining weights by the round of IMP they get removed during, and plot the mean. This process is repeated for all subsequent rounds of IMP. Because the number of weights remaining after each round of IMP decreases, the removal of any given surviving weight leads to an increasingly large impact on the preactivations. To avoid seeing trivial differences in scaling, we report $N_W(n) \cdot \text{cavity}(W_{ij})$, where $N_W(n) = N(1-s)^n$ is the number of weights remaining after $n$ rounds of IMP. This acts as a form of normalization to enable more clear visualization, but it does not affect our primary conclusion, which rests on the sign of the cavity score.

We find a systematic pattern, where the weights that get removed in IMP round $n$ have a positive mean cavity score after $n-1$ rounds of pruning, and all weights that get removed in later rounds of IMP have a negative mean cavity score (Fig. 5C). A striking example is found for the weights that get removed at IMP round 8 (Fig. 5C, vertical gray dashed line). Their mean cavity score remains well below zero ($\text{cavity}(W_{ij}) < -0.1$), when evaluated after IMP round $n = 0, 1, ..., 6$ (red to purple lines that intersect the gray dashed line – Fig. 5C). However, after seven rounds of IMP ($n = 7$), their cavity score suddenly becomes positive (blue line that intersects the gray dashed line – Fig. 5C). This indicates that, for the first time, their removal is expected to lead to an increase in the non-Gaussian statistics. Remarkably, *this is precisely when they are removed*.

That we see this trend for all rounds of IMP strongly supports the idea that it is the *order*, in addition to the identity, of the weights that IMP removes, that leads to an increase in the non-Gaussian preactivation statistics. Furthermore, the order maximizes the amount of non-Gaussanity of the FCN representation, as IMP does not, on average, remove weights when their pruning would increase the Gaussian statistics of the preactivations. This provides strong support for our central hypothesis, $\mathbf{H}^*$.

## 4. Discussion

Motivated by the finding that iterative magnitude pruning discovers local receptive fields in fully connected neural networks [17], we sought to investigate the cause of this phenomenon. By connecting this observation of Pellegrini and Biroli [17] with recent work by Ingrosso and Goldt [26], who demonstrated that local RFs can emerge in FCNs trained on inputs with sufficiently strong non-Gaussian statistics, we hypothesized that IMP iteratively maximizes the the amount of non-Gaussian statistics present in the FCN preactivation, at each round of pruning (Hypothesis $\mathbf{H}^*$). While such a greedy algorithm may not, in general, be optimal, by creating a feedback loop (where increased non-Gaussianity leads to greater localization, which leads to greater non-Gaussianity), such a strategy could lead to strong localization.

To support this hypothesis, we made three observations. First, we showed that non-Gaussian statistics are indeed *necessary* for local RFs to emerge via IMP, as training FCNs on datasets that match ImageNet32 in the first two cumulants, and have zero higher-order cumulants, leads to IMP finding masks with no localization (Fig. 3). Second, we found that IMP amplifies the amount of non-Gaussanity present in the FCN representation with each round of pruning, making the distribution of preactivations significantly more non-Gaussian than oneshot pruning (Fig. 4). Third and finally, we develop a method to measure the impact of removing individual weights, inspired by the "cavity method" of statistical physics, to provide evidence that the order in which IMP removes FCN weights is aligned with when their removal would increase the non-Gaussanity of the preactivations (Fig. 5). Collectively, these results provide strong evidence for $\mathbf{H}^*$.

**Limitations.** This work is focused on shallow FCNs, trained on a down-sampled version of ImageNet (ImageNet32 [53]). While this setting is removed from many modern implementations of machine learning (ML), we believe this is justified by the fact that it is the *only* setting in which the

inductive bias discovered by IMP is well characterized [17]. To fully prove our central hypothesis ($\mathbf{H}^*$), it is necessary to show that the weights IMP removes are optimal in increasing the preactivation kurtosis, over all possible choices of pruning. This is infeasible. However, our cavity method (Sec. 3.3) is able to provide evidence of a first order approximation of optimality of IMP (Fig. 5). We believe this, along with the rest of our results, provide evidence for the validity of $\mathbf{H}^*$.

**Future directions.**  Recent work examining the training dynamics that drive local RF emergence in a simplified batch gradient setting, with synthetic data and a single hidden layer FCN, argued that training on inputs with kurtosis $< 3$ (negative excess kurtosis) is essential for localization [27]. This is in contrast to our results on FCNs trained on ImageNet32, where we found localization in pruning masks (Fig. 1), despite the preactivation kurtosis being $> 3$ (Fig. 4B). This difference may point to implicit properties of stochastic gradient descent and its interaction with large-scale computer vision datasets, which future work can explore. That the order in cavity score is well aligned to the order of when weights are removed via IMP, *before* any pruning occurs (Fig. 5C – IMP round 0 line), suggests that it could be used to more efficiently find localized RFs. Future work can explore how to best use this metric as a cost function to develop new pruning algorithms.

**Implications for sparse ML.**  In addition to IMP, other approaches have been developed to identify sparse subnetworks. These approaches include methods that oneshot prune dense models based on gradients and Hessians of the loss [56–58], and dynamic sparsity methods that optimize over masks [59, 60], thereby never training on the dense model. Our results suggest that the ability of IMP to identify local RFs in FCNs comes both from its iterative nature and its leveraging of the dense model's training dynamics. This may explain why Pellegrini and Biroli [17] found that SNIP [57] and SynFlow [58] were unable to find as localized RFs as IMP. Additionally, this calls into question whether dynamic sparsity training approaches are capable of identifying local RFs. More generally, our results have implications on the ability of these other sparsity algorithms to extract and maintain strong inductive biases.

**Beyond FCNs.**  While FCNs offer a tractable framework with which to study the impact of IMP on internal representations, the success of IMP on modern architectures, such as CNNs [3, 6, 12], suggests that IMP may similarly be increasing the non-Gaussian statistics of the preactivations and driving the emergence of even stronger local RFs. Future work can explore the effect of IMP on the non-Gaussian statistics in CNNs. IMP has also been successfully applied in settings beyond computer vision classification tasks, including natural language processing [11, 38] and reinforcement learning [38, 42]. In such settings, useful inductive biases and their interaction with the statistical properties of the inputs are less clear. An exciting avenue of future work would be to leverage the cavity method developed in this work to explore the impact of individual weights and the order with which IMP removes them. Work in this direction may shed additional light on why IMP can struggle to sparsify large language models [61, 62].

Finally, our results also provide a possible explanation for the ability of IMP derived subnetworks ("winning tickets") to transfer across computer vision tasks [6–13] and CNN architectures [14]. In particular, if IMP increases the preactivation kurtosis in CNNs, then winning tickets will have amplified non-Gaussian statistics, which are broadly useful in computer vision tasks.

# Acknowledgements

SG gratefully acknowledges funding from the European Research Council (ERC) for the project "beyond2", ID 101166056; from the European Union–NextGenerationEU, in the framework of the PRIN Project SELF-MADE (code 2022E3WYTY – CUP G53D23000780001), and from Next Generation EU, in the context of the National Recovery and Resilience Plan, Investment PE1 – Project FAIR "Future Artificial Intelligence Research" (CUP G53C22000440006).

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

# A. Measuring localization

A method for quantifying the localization of the RFs that emerges from IMP and oneshot pruning has not been developed. As a simple approximation, we choose to fit the correlation function, $S_i(d, n)$ (Eq. 1), with a two-dimensional Gaussian. This was motivated by the fact that Pellegrini and Biroli [17] found that the correlation function, averaged across all the most connected IMP masks, had Gaussian-like structure. As illustrated in Fig. S1A, the correlation function associated with individual IMP masks can also be well approximated by two-dimensional Gaussians. This is true even when the underlying mask does not appear to be well described by a Gaussian, thus emphasizing the rationale for fitting using $S_i(d, n)$. In addition, the mean-squared error (MSE) associated with the Gaussian fit of the IMP masks was significantly smaller than the MSE associated with the Gaussian fit of random pruning masks (Fig. S1B). We note that, while we believe this is evidence for the appropriateness of our method to quantify localization (via the standard deviation of the fit Gaussian), our larger point of IMP selectively increasing preactivation kurtosis is not dependent on these results.

# B. PCA and ICA Experiments

We perform principal component analysis (PCA) and independent component analysis (ICA) [52] using `sklearn.decomposition.PCA` and `sklearn.decomposition.FastICA`, setting the number of components to $n = 64$. PCA and ICA were applied to all $50,000$ test images in the ImageNet32 dataset. The results are shown in Fig. 2.

Given that the IMP masks and ICA components are localized, we investigate whether they overlap in the features they extract. To do this, we compute the cosine similarity between every ICA component and each IMP mask. We denote the largest cosine similarity, across all 64 ICA components, as the "cosine similarity" between the IMP mask and the ICA components. Example matches (with the mean cosine similarity scores) are shown in Fig. S2A. We note that this approach is limited. For instance, there may be transformations (e.g., rotations) that could make the IMP masks more aligned to the ICA components. However, we believe this is still a valuable approximation of the similarity between the ICA and IMP. Overall, the distribution of cosine similarity is skewed towards small values (Fig. S2B), indicating that ICA and IMP extract largely distinct features.

# C. ImageNet32 Experiments

## C.1. ImageNet32-GP

To construct the Gaussian clone of ImageNet32 [53], we utilize the same approach previously used to create clones of CIFAR-10 [28]. Namely, we sampled from two-dimensional Gaussian distributions that were fit to 100,000 images from the ImageNet32 dataset. Each color channel and image class had its own fit. This led to an average of 100 images per class used for the fit. While not a small number, we recognize that this is a limitation. With the two-dimensional Gaussian fits, we sampled the same number of images as the original ImageNet32: 1,281,167 train images and 50,000 test images. Because only $\approx 8\%$ of the images were used to generate the model, the distribution of number images per class is not guaranteed to match the true distribution. Because we use the Gaussian clone to assess the emergence of local RFs, and not the performance capability of the FCN models on ImageNet32-GP, we do not believe this difference affects our primary conclusion.

## C.2. FCN model

For our FCN experiments, we follow the approach of Pellegrini and Biroli [17]. We use the official implementation publicly available[3], with three minor modifications to reduce computational costs. First, we scale the FCN down to have two hidden layers, instead of three. Given that Pellegrini

---

[3] `https://github.com/phiandark/SiftingFeatures`

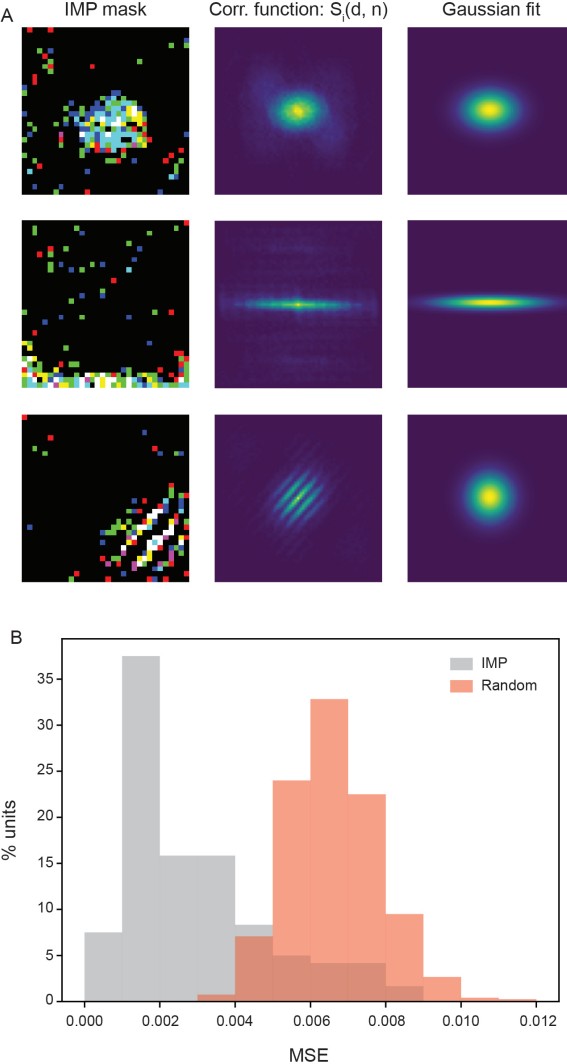

Figure S1: **Measuring localization of RFs with two-dimensional Gaussians.** (A) Three example IMP masks (after 10 rounds of IMP) (left column), with their associated correlation function $S_i(d, n)$ (middle column). The Gaussian fits (right column) demonstrate strong qualitative agreement with the true $S_i(d, n)$. (B) Distribution of mean-squared error (MSE) associated with the Gaussian fits of IMP and random pruning masks. The random pruning masks serve as a baseline, as we expect them to not have much Gaussian structure in their correlation functions.

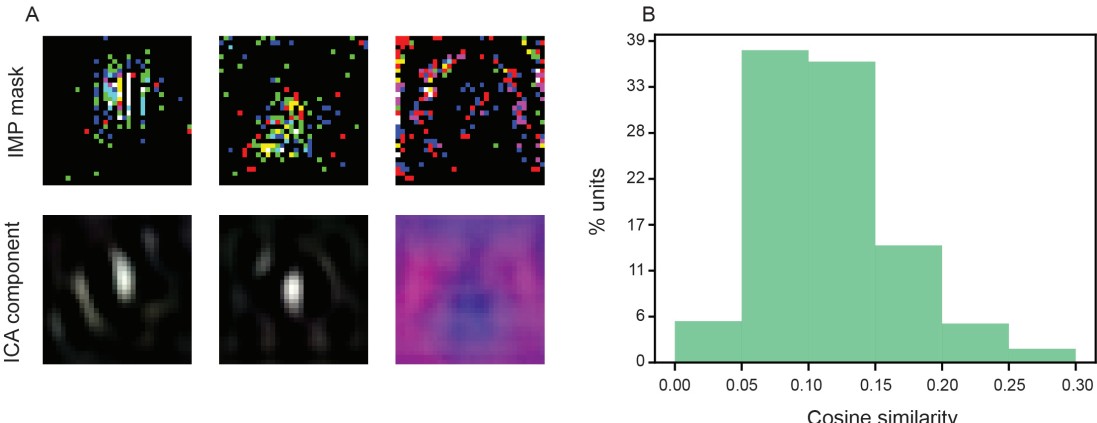

Figure S2: **IMP and ICA find different local features.** (A) Example IMP masks (top row) and the ICA component they have the highest cosine similarity with (bottom row). Masks are from three independent seeds, each chosen as having the mean cosine similarity across the population of masks considered. (B) Distribution of cosine similarity between IMP and ICA across three independent seeds.

and Biroli [17] found similar localization of IMP masks when using FCNs of greater widths, we reasoned that the width of the FCN does not play a major role in the emergence of local RFs. Our results (Fig. 1) demonstrate that IMP finds local RFs, even in this shallower FCN architecture. Second, we decrease the batch size from 1000 to 40. Given that a larger learning rate to batch size ratio has been found to lead SGD to converge to more generalizable solutions [63], we reasoned that – if anything – this choice would lead to greater localization (as local RFs are a generalizable computation). Lastly, we decreased the total number of training iterations from 100,000 to 40,000. Given that Pellegrini and Biroli [17] found similar (but slightly weaker) results when training for 10,000 training steps, we reasoned that 40,000 training steps would be a good intermediary. For details on all other architecture and training hyper-parameters, see Table S1.

Table S1: FCN parameters for ImageNet32 experiments.

| Parameters | |
| --- | --- |
| Architecture | 3072:1024:1024:1000 |
| Activation function | ReLU |
| Batch size | 40 |
| Learning rate | 0.1 |
| Optimizer | SGD |
| Training iterations | 40,000 |
| Rewind iteration ($t_{\text{rewind}}$) | 1,000 |

## D. Second layer preactivation kurtosis

To test whether IMP increases the non-Gaussianity of later FCN layers, we compute the preactivation kurtosis of the second hidden layer, as a function of IMP round (as noted in Appendix C.2, our FCN architecture only contains two hidden layers). Note that in this case, the preactivation equation (Eq. 3) is given by

$$\lambda_i(X) = \sum_{j=1}^{N} W_{ij} \tilde{a}_j(X) + b_i, \tag{7}$$

where $\tilde{a}_j(X)$ is the activation of the $j^{\text{th}}$ hidden unit in the first hidden layer (Eq. 2), with batch normalization. That is, $\tilde{a}_j(X) = \gamma_j a_j(X) + \beta_j$, where $\gamma_j$ and $\beta_j$ are the learned parameters for the second layer.

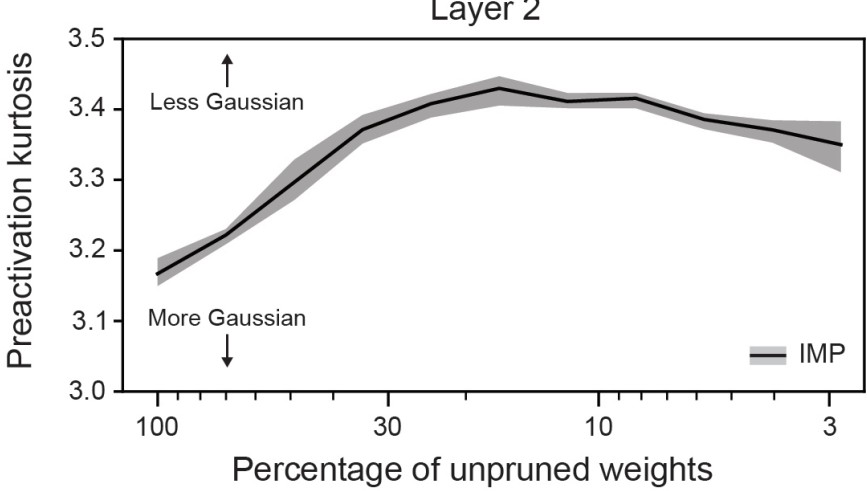

Figure S3: **Preactivation kurtosis increases with IMP.** Same as Fig. 4B, but when computing the preactivation kurtosis of the FCN's second layer.

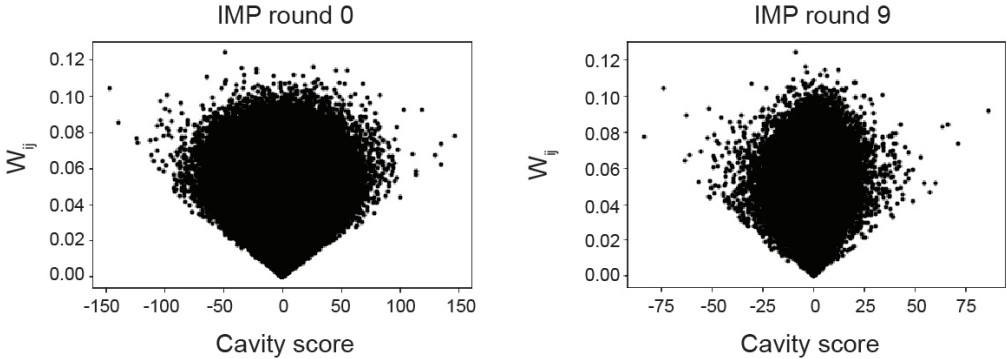

Figure S4: **Cavity score is not strongly connected to parameter value** (A)–(B) Parameter value, as a function of cavity score, for IMP round 0 and 9. Each dot corresponds to one non-pruned weight.

As with the preactivations of the first hidden layer, we find that kurtosis increases with rounds of IMP (Fig. S3), plateauing earlier than the preactivation kurtosis of the first layer (Fig. 4B). The preactivation kurtosis slightly decreases at higher rounds of IMP (Fig. S3). This difference may be due to the fact that the second layer units are getting inputs from the hidden units in the first layer, as opposed to inputs from the images. Future work can explore the localization in later layers, and how it compares to localization in the first hidden layer, in more detail.

