# OpenReview forum: "How Iterative Magnitude Pruning Discovers Local Receptive Fields in Fully Connected Neural Networks"
_CPAL.cc/2025/Proceedings_Track — CPAL 2025 (Proceedings Track) Poster_

### Official Review · Reviewer_F7z5 · 2025-01-10

**Rating:** 7
**Confidence:** 3

**Review:**

This paper investigates the effects of iterative magnitude pruning (IMP) on fully connected neural networks. The paper advances the hypothesis that IMP maximally increases the non-Gaussianity of its internal representations, and finds supporting evidence for this hypothesis. Overall, the paper makes clear claims with supporting evidence and has good motivation and structure. I found it to be interesting to read and learned from it, so I think it could be a worthy inclusion to this conference.

The technical strengths of the paper, in my view, lie primarily in the experiments conducted in Sections 3.2-3. I’m less sure what to conclude about the localized receptive fields. I go into more detail about these questions/concerns below.

Questions and suggestions for the authors:

1. I think the hypothesis (line 48) could be made more precise. Could non-Gaussian be replaced with kurtosis? “statistics represented by the network” also seems strange — should this not be “statistics of the pre-activations” (or other specific features of the network, if not these)?

2. For Figure 2A, there are many features that appear to be neither “localized” (143) nor “resemble Gabor filters” (144). The authors should amend this claim and it would be helpful to explain why this is the case.

3. Can the authors justify or explain why two-dimensional Gaussian fits are an appropriate framework for quantifying the receptive field size? It seems that in many cases (e.g. Fig. 3B) that the masks are not well fit by a 2D Gaussian, which makes it difficult to determine what the RF width should mean. I am also puzzled by the small deviations in RF width shown in Fig. 4C, especially for ImageNet32-GP. Given that the bars show “minimum and maximum of three” networks, is it surprising that all RF widths are essentially the same?

4. It seems that when only a small percentage of weights are pruned, IMP receptive fields are slightly larger than one-shot (Fig. 4A). Why is this, and again is it surprising that the variance is so low?

5. The “cavity score” experiments in Section 3.5 provide evidence that weights are pruned when removing them would increase pre-activation kurtosis. This appears to be a greedy algorithm - can the authors comment on whether or why this would be optimal for the end goal of increasing non-Gaussianity? I’d like to see why this approach would achieve maximal non-Gaussianity (the hypothesis) rather than merely be an approximation to the maximum.

6. Finally, I’d be curious to see a scatterplot of the relationship between weight magnitudes and cavity score (possibly for different IMP rounds), in case the authors have this data available.

Minor points:

- 140: “principle” -> “principal”
- line 266: “work to with” -> “work to” or “work with”

---

### Official Review · Reviewer_foVF · 2025-01-11
**Cool paper**

**Rating:** 7
**Confidence:** 3

**Review:**

Summary:

The author provide explain why applying iterative magnitude pruning (IMP) leads to the emergence of local receptive fields in neural networks. The author first hypothesize that IMP discovers local RFs in FCNs by maximally increasing the non-Gaussian statistics represented by the network. And they provide many experiments to verify this hypothesize.

Strength:
The paper is well written and the experiments are all on points.
The topic of the paper on IMP and locality is well aligned with the purpose of the conference.

Weakness:
Honestly, I can't think of any weakness. Some insight on how the result could help understanding IMP would be nice, but could also be a future direction.

Question:
What do the author think the broader impact of the result? As mentioned in the introduction, the idea of non-guassian statistic, sparsity, locality in natural image is well explored in the literature of computational neuroscience and early computer vision research before deep learning. I think the interesting direction would be that if the result from this paper could help us understand the mechanism of "IMP" better. If locality emerge from increasing non-Gaussian statistics in the neural network, then why IMP encouraging increases the non-Gaussian statistics. The author mentioned in the introduction that the paper focus on the emergence of local receptive field without explicit regularization of the model. But the idea of IMP seems like a sparsity constraint/regularization on the parameters of model. It makes sense that increasing sparsity on the parameters would promote locality, because to have local receptive field, the receptive field needs to be sparse in the first place.

---

### Official Review · Reviewer_FrHy · 2025-01-20
**Review for paper 7**

**Rating:** 6
**Confidence:** 3

**Review:**

## Summary
This paper provides substantial empirical evidence and interpretable metrics to understand the interplay between the iterative nature of IMP, data statistics, and emerging localized structure in the pruned subnetworks.

---

## Evaluation
- **Quality**:
The quality of the work is high. The theoretical background linking local receptive fields to non-Gaussian statistics is clearly motivated by prior literature on independent component analysis (ICA). The paper’s experimental design—comparing ImageNet32 to Gaussian clones, and comparing iterative pruning to oneshot pruning—strongly supports the claim that non-Gaussianity drives the emergent inductive bias. The introduction of a “cavity method” to evaluate the effect of individual weight pruning is a novel and illuminating technique.

- **Clarity**: The paper is mostly clear in its exposition. Key concepts—kurtosis, higher-order cumulants, and how these relate to localized features—are well introduced. The step-by-step analysis (importance of non-Gaussian statistics, measuring representation kurtosis, cavity method for weights) is logically structured and easy to follow.

---

## Pros and Cons

### Pros
- **Clear Empirical Evidence**: The paper convincingly shows that local RFs do not emerge under Gaussian-cloned data, suggesting higher-order statistics are necessary.
- **Iterative vs. Oneshot Pruning**: The comparison highlights how iterative pruning consistently increases kurtosis, validating the need for multiple rounds of pruning rather than a single prune.
- **“Cavity Method”**: This metric is a compelling idea that provides weight-level insights into how pruning affects preactivation statistics.
- **Strong Theoretical and Biological Motivation**: The link to ICA and visual cortex motivates why local RFs correspond to non-Gaussian signals.

### Cons

- **Scope of Experiments**: The work remains focused on shallow fully connected networks and ImageNet32. Future validation on deeper networks or different data modalities would strengthen the claims of generality.
- **Optimality Claim**: While the cavity analysis strongly suggests that IMP removes weights in an order that maximally boosts non-Gaussian statistics, a strict proof of optimality (given the combinatorial explosion of possibilities) is outside the scope.
- **Complex Interaction with SGD**: The paper attributes the final shape of the internal representations to pruning, but training dynamics (beyond second-order effects) may also contribute significantly.
Limited Discussion on Practical Extensions: Ideas on how to scale or leverage the cavity method in large-scale architectures (e.g., CNNs, transformers) could be elaborated.

---

## Suggestions

### Further Exploration of Different Layers
The experiments focus on the first hidden layer. It would be interesting to see how deeper hidden layers’ representations evolve under iterative pruning. Does the same non-Gaussian amplification hold layer by layer?
### Implications for Modern Architectures
Although the authors briefly discuss the possibility of local RFs in CNNs, providing preliminary experiments or theoretical arguments for deeper networks would broaden the scope of these findings.
### “Cavity Method” Scalability
The cavity metric requires measuring the preactivation distribution repeatedly, which might become expensive in very large networks. Discussion on how to approximate or optimize it would be valuable.
### Connection to Other Pruning Methods
The paper’s focus is on magnitude pruning. It would be insightful to contrast these findings with gradient-based oneshot pruning or dynamic sparse training methods, to see whether the iterative, rank-ordered removal of weights still drives these strong inductive biases.
### Link Between Kurtosis and Edge-Like Filters
The paper rightly shows that non-Gaussian features can correspond to edges. It could be illustrative to compare the actual filters discovered (visualized as weight patterns) to the Gabor-like filters from ICA, clarifying how close they are in structure.

---
Overall, this paper is good: it is clear, well-executed, and addresses a gap in the understanding of how iterative pruning fosters beneficial inductive biases in FCNs. The introduction of the “cavity method” and the link to non-Gaussian statistics enhance its originality. Although limited to smaller networks and images, the insights are promising.

---

### Meta-Review · Area_Chair_WwgE · 2025-02-04

**Recommendation:** Accept (Poster)
**Confidence:** 4

**Metareview:**

This paper provides empirical support for how iterative magnitude pruning helps fully connected networks discover local receptive fields in their weights. The experiments show that local RFs do not emerge without non-gaussian statistics and that IMP (maximally) increases the non-gaussian statistics of FCN representations.

**Why accept:** The paper is of sufficient interest to the conference audience. All reviewers felt that the experiments were sufficiently interesting and have provided high scores. I also believe the experiments demonstrate the papers' claims, and recommend acceptance.

**Why not a higher rating:** While the paper is correct and interesting, it is quite narrow in the question it studies. In its current form, it does not provide recommendations for using/not using IMP, using/not using regularization to discover these local RFs, how the findings will go beyond image datasets.

The reviewers have raised the following recommendations to properly scope the paper's claims (and I agree that these should be addressed):

1. Explain why two-dimensional Gaussian fits are an appropriate framework for quantifying the receptive field size
2. Tone down the overstatement that ICA features are localized on the ImageNet32 dataset
3. Clearly explain how features discovered by IMP may transfer - also how the statistics of IMP masks are tied to the statistics of the data
4. Compare the masks found with IMP to ICA features
5. Include experiments and findings on the second layer of FCNs in the paper.

---

### Decision · Program_Chairs · 2025-02-11

Accept (Poster)